# Single-Cell Analysis Differentiates the Effects of p53 Mutation and p53 Loss on Cell Compositions of Oncogenic Kras-Driven Pancreatic Cancer

**DOI:** 10.3390/cells12222614

**Published:** 2023-11-12

**Authors:** Xinlei Sun, Daowei Yang, Yang Chen

**Affiliations:** 1Department of Translational Molecular Pathology, The University of Texas MD Anderson Cancer Center, Houston, TX 77030, USA; 2Sheikh Ahmed Center for Pancreatic Cancer Research, The University of Texas MD Anderson Cancer Center, Houston, TX 77030, USA

**Keywords:** pancreatic cancer, tumor microenvironment, single-cell analysis

## Abstract

Pancreatic ductal adenocarcinoma (PDAC) is a devastating malignant disease with a dismal prognosis. In the past decades, a plethora of genetically engineered mouse models (GEMMs) with autochthonous pancreatic tumor development have greatly facilitated studies of pancreatic cancer. Commonly used GEMMs of PDAC often harbor the oncogenic KRAS driver mutation (*Kras^G12D^*), in combination with either p53 mutation by knock-in strategy (*Trp53^R172H^*) or p53 loss by conditional knockout (*Trp53^cKO^*) strategy, in pancreatic cell lineages. However, the systematic comparison of the tumor microenvironment between *Kras^G12D^*; *Trp53^R172H^* (KP^mut^) mouse models and *Kras^G12D^*; *Trp53^cKO^* (KP^loss^) mouse models is still lacking. In this study, we conducted cross-dataset single-cell RNA-sequencing (scRNA-seq) analyses to compare the pancreatic tumor microenvironment from KP^mut^ mouse models and KP^loss^ mouse models, especially focusing on the cell compositions and transcriptomic phenotypes of major cell types including cancer cells, B cells, T cells, granulocytes, myeloid cells, cancer-associated fibroblasts, and endothelial cells. We identified the similarities and differences between KP^mut^ and KP^loss^ mouse models, revealing the effects of p53 mutation and p53 loss on oncogenic KRAS-driven pancreatic tumor progression.

## 1. Introduction

Pancreatic ductal adenocarcinoma (PDAC) is one of the deadliest cancer types and is resistant to therapies [1]. Unfortunately, most patients already present with either locally advanced or metastatic PDAC upon diagnosis, thus preventing the surgical resection of the tumors [2]. PDAC is characterized by a desmoplastic microenvironment comprising various cell populations, including cancer cells, cancer-associated fibroblasts (CAFs), endothelial cells, and immune cells [3]. Recent observations utilizing a single-cell RNA-sequencing (scRNA-seq) technique have shed light on the landscape of these cell populations in the tumor microenvironment of PDAC [4,5,6,7,8,9,10,11,12]. Our recent studies using scRNA-seq and various transgenic mouse models of PDAC have identified the changes in specific cell populations associated with different tumor stages and genotypes [10,11,13]. The development of PDAC is frequently associated with genetic alterations of KRAS (*KRAS* gene) and p53 (*TP53* gene). While oncogenic KRAS alteration is dominated by the G12* (especially G12D) mutation as the key driver of PDAC, p53 mutations can have either gain-of-function or loss-of-function effects [14]. One of the most studied p53 mutations in the context of PDAC mouse models is p53^R172H^, corresponding to the human p53^R175H^. The p53^R172H^ mutation has been shown to be associated with p53 gain of function and overexpression in cancer cells, leading to oncogenic effects [15,16,17,18]. The other commonly observed p53 mutations in human cancers include R248Q, R248W, R273C, R273H, G245S, H179R, Y163C, Y220C, Y234C, G249, and R282, which are associated with p53 gain-of-function mutations [19,20,21]. Recent studies reported that p53^R248W^ can promote the metastasis of human pancreatic cancer [22]. In this study, we focused on the p53^R172H^ as a representative p53 gain-of-function mutation (hereafter referred to as “p53 mutation”), as compared to p53 deletion or loss-of-function mutations (hereafter referred to as “p53 loss”). Previous studies using various transgenic mouse models have shown that gain-of-function p53 mutations can have significantly different oncogenic and pro-metastatic effects as compared to the loss-of-function p53 mutations in the context of PDAC [15,16]. However, it remains unclear whether p53 mutation and p53 loss can differentially influence the various cell populations within the tumor ecosystem. Here, we performed cross-dataset analyses of scRNA-seq data from PDAC samples to systematically reveal the differences between p53 mutation and p53 loss in regulating the tumor microenvironment of oncogenic KRAS-driven PDAC.

## 2. Methods

### 2.1. Single-Cell RNA-Sequencing (scRNA-Seq) Analysis

The datasets used in this study were obtained by downloading our previously deposited files from the Gene Expression Omnibus (GEO) repository as described in our recent studies [10,11]. Specifically, the scRNA-seq data of early and late primary tumors from a KP^mut^ (*Kras^G12D^*; *Trp53^R172H^*) mouse model with KRAS^G12D^ mutation and p53^R172H^ mutation were from the GSE198815 dataset, and the scRNA-seq data of late primary tumors from a KP^loss^ (*Kras^G12D^*; *Trp53^cKO^*) mouse model with KRAS^G12D^ mutation and p53 loss mice were from the GSE166298 dataset. As previously described, early-stage PDAC was defined by having less than 10% pancreatic adenocarcinoma areas, while late-stage PDAC was defined by having greater than 50% pancreatic adenocarcinoma areas. All the scRNA-seq datasets were derived from primary pancreatic tumors of background-matched mice. The KP^mut^-Early (early-stage primary tumor with KRAS and p53 mutations) group contained 6175 cells from 3 individual early-stage KP^mut^ mice. The KP^mut^-Late (late-stage primary tumor with KRAS and p53 mutations) group contained 9479 cells from 3 individual late-stage KP^mut^ mice. The KP^loss^-Late (late-stage primary tumor with KRAS mutation and p53 loss) group contained 4072 cells derived from 3 individual late-stage KP^loss^ mice. All samples were processed using the same protocol by the Sequencing and Microarray Facility at MD Anderson Cancer Center (MDACC). To analyze the cells, we used the 10X Genomics Chromium controller and Single Cell 3’ Reagent Kits v2 (10X Genomics, Pleasanton, CA, USA). This generated cDNA, which was then amplified to create Illumina sequencing libraries. We used the Illumina NextSeq 500 to sequence the libraries, with a run format of 26 cycles for read 1, 8 cycles for index 1, and 124 cycles for read 2. For data analysis, we installed the R packages ‘Seurat’ version 4.4.0, ‘dplyr’, and ‘cowplot’ (version 4.3.1). We utilized these tools to filter low-quality cells based on a minimum of 200 and a maximum of 7000 genes per cell and removed cells with more than 10% of the mitochondrial genome. The “RunUMAP” function was used to cluster the cells, with the principal components determined by the “JackStrawPlot” and “ElbowPlot” functions. Cluster signature genes were identified using the “FindAllMarkers” function, and differentially expressed genes were identified for selected cell clusters between different groups using the “FindMarkers” function. The “DoHeatmap” function was used to display the top signature genes in each cluster, while “VlnPlot” and “DotPlot” functions were used to examine the expression profiles of selected genes across cell clusters. Relative cell type abundance was visualized with the R package ‘ggplot2’ (version 3.4.3).

### 2.2. Gene Signatures

Inflammation-signature-associated genes, including *Ifng*, *Ifngr1*, *Ifngr2*, *Il10*, Il12a, *Il12b*, *Il12rb1*, *Il12rb2*, *Il13*, *Il17a*, *Il18*, *Il18r1*, *Il1a*, *Il1b*, *Il2*, *Il21*, *Il21r*, *Il22*, *Il23a*, *Il23r*, *Il2rg*, *Il4*, *Il5*, *Il6*, *Jun*, *Rela*, *Rora*, *Rorc*, *S100a8*, *S100a9*, *Stat1*, *Stat3*, *Stat4*, *Stat6*, *Tgfb1*, *Tgfb2*, *Tgfb3*, and *Tnf*, were obtained from a previous publication [23].

Neutrophil-related functions, including immunosuppression (*Havcr2*, *Fcgr2b*, *Il4ra*, *Cd274*, *Hif1a*), ECM remodeling (*Adamdec1*, *Ctsc*, *Ctsb*, *Rgcc*, *Ctss*, *Ctsz*, *Adam17*, *Adam10*, *Adam8*), tumor proliferation (*Tgfb1*, *Tnf*, *Il1a*), myeloid cell recruitment (*Ccl3*, *Mif*, *Cxcl14*, *Csf1*, *Vegfa*, *Ccl4*, *Cxcl3*), angiogenesis (*Vegfa*, *Snd1*, *Mtdh*, *Itga5*, *Tnf*, *Cxcl3*, *Anxa3*, *Hmgb1*, *Hif1a*, *Sema4d*, *Lrg1*, *Chil1*), neutrophil degranulation (*Abr*, *Anxa3*, *Cd177*, *Itgam*, *Itgb2*, *Itgb2l*, *Pikfyve*, *Pram1*, *Ptafr*, *Spi1*, *Stx11*, *Syk*), neutrophil cytotoxicity (*Ncf1*, *Myd88*, *Trem3*, *Trem1*, *Tusc2*, *Cybb*, *Cybc1*, *Ncf2*, *Ncf4*, *Rac1*, *Rac2*), interferon signaling (*Adar*, *Isg15*, *Isg20*, *Rsad2*, *Ifit1*, *Ifit3*, *Ifitm1*, *Ifitm3*, *Irak1*, *Oas3*, *Stat1*, *Stat2*, *Irf7*, *Cxcl10*), were obtained from a previous publication [24]. All obtained calculation results are visualized through the “ggboxplot” function.

### 2.3. Gene Set Enrichment Analysis (GSEA)

We utilized the “Wilcoxauc” function to compute cell groups and obtained gene markers for each group. These markers were then compared using the HALLMARK category from the Molecular Signatures Database (MSigDB) gene sets. Following this, we conducted GSEA analysis using the package ‘fgsea’ (version 1.26.0), and the outcomes were presented using ‘ggplot2’ (version 3.4.3), pathway heatmaps, and correlations visualized using the ‘dittoSeq’ package (version 1.12.1).

### 2.4. The Cancer Genome Atlas (TCGA) Data

We obtained TCGA data of a pancreatic cancer cohort from the cBioPortal database and identified cases with *TP53* truncating mutations as p53 loss (loss-of-function) in the context of KRAS G12* mutations. In comparison, we also identified cases with *TP53* mutations including R175H, R248Q, R248W, R273C, R273H, G245S, H179R, Y163C, Y220C, Y234C, G249, and R282 as p53 gain-of-function mutations [19,20] in the context of the KRAS G12* mutations. We then compared the mRNA expression levels of selected genes between p53 loss and p53 mutation groups. The heatmaps was visualized using the ‘pheatmap’ package (version 1.0.12) after standardization and “ward.D2” clustering_method treatment. In addition, the above data together with patient survival data were analyzed using the log-rank (Mantel–Cox) test.

### 2.5. Statistical Analysis

Statistical analyses of mRNA expression between p53 loss and p53 mutation were performed with unpaired, one-tailed *t*-test using GraphPad Prism (GraphPad Software, Prism 10). The expression data of indicated genes among TCGA pancreatic adenocarcinoma patient samples were based on the RNA Seq V2 RSEM data. TCGA data were downloaded from the cBioPortal database [25,26]. A *p* value < 0.05 was considered statistically significant. Error bars represent standard error of the mean (S.E.M.).

## 3. Results

### 3.1. Single-Cell Analysis Reveals Unique Differences in the Composition and Genomic Profile of Cell Populations between the KP^mut^ and KP^loss^ Pancreatic Tumors

First, we included our recently published single-cell RNA-sequencing (scRNA-seq) datasets [10,11] to compare two mouse models with autochthonous pancreatic ductal adenocarcinoma (PDAC): the KP^mut^ (*Kras^G12D^*; *Trp53^R172H^*) mouse model with KRAS^G12D^ mutation and p53^R172H^ mutation, as compared to the KP^loss^ (*Kras^G12D^*; *Trp53^cKO^*) mouse model with KRAS^G12D^ mutation and p53 loss (Figure 1A,B). We then compared the primary tumors across various cell types from early-stage KPC mice (KP^mut^-Early), late-stage KPC mice (KP^mut^-Late), and late-stage KPPF mice (KP^loss^-Late) (Figure 1C). Our analysis revealed significant differences in the composition of major cell populations between the KP^mut^-Early, KP^mut^-Late, and KP^loss^-Late (Figure 1D,E). As PDAC progressed in the contexts of different p53 alterations, the composition of cancer cell subpopulations significantly altered. Notably, a unique cancer cell subtype was present in the tumors of KP^mut^-Late mice but not KP^loss^-Late mice (Figure 1C). Cell number and phenotype alterations in neutrophils, fibroblasts, and other cell types were also observed (Figure 1C–E). T and B cell numbers were relatively high in KP^mut^-Early tumors and significantly decreased in KP^mut^-Late and KP^loss^-Late tumors (Figure 1D,E). Myeloid cell numbers in KP^mut^-Late and KP^loss^-Late were far beyond the KP^mut^-Early (Figure 1D,E). These results indicate that decreased lymphocyte number and increased myeloid cell number are associated with late-stage PDAC regardless of p53 alteration subtypes. Additionally, we utilized a recently reported methodology of analyzing the inflammation-related signals [23] and observed a significant increase in inflammation scores in KP^mut^-Late and KP^loss^-Late samples (Figure 1F), indicating the enrichment of inflammation-related signals in late-stage tumors but not in early-stage tumors.

### 3.2. KP^mut^-Late Mice Harbor a Unique Cancer Cell Subcluster as Compared to KP^loss^-Late Mice (or KP^mut^-Early Mice)

Based on our analysis, we have identified three distinct subclusters of cancer cells, namely ‘Cancer cell 1’, ‘Cancer cell 2’, and ‘Cancer cell 3’, as shown in Figure 2A. Specifically, Cancer cell 1 was present in all three groups, as the predominant cancer cell subtype in KP^mut^-Early and KP^loss^-Late tumors (Figure 2B). Cancer cell 2 was mainly enriched in KP^mut^-Late tumors, whereas Cancer cell 3 was mainly enriched in KP^mut^-Early tumors (Figure 2B). To further investigate the similarities and differences between these three subpopulations of cancer cells, we analyzed the marker genes of these cancer cell subtypes. As expected, all three subtypes of cancer cells expressed the generic pancreatic cancer cell marker genes, such as *Krt8* and *Krt18* (Figure 2C,D). Specifically, Cancer cell 1 expressed *Gcnt3*, *Mmp7*, and *Vsig2*, which are associated with the epithelial/classical subtype of pancreatic cancer cells [27,28,29] (Figure 2C,D). Cancer cell 2 expressed *Sprr1a*, *Lgals7*, and *Ifitm1*, which are associated with the mesenchymal/basal-like subtype and a poor prognosis in PDAC [30,31,32]. Cancer cell 3 represented the early-stage cancer-initiating cells with the expression of acinar cell genes such as *Clps*, *Try4*, and *Try5* (Figure 2C,D), as well as epithelial marker genes such as *Epcam* and *Cdh1* (Appendix A).

To establish a correlation between cancer cell subclusters and PDAC subtypes [32], we analyzed the expression profiles of signature genes associated with the classical and basal-like subtypes (Figure 2E, Appendix A). In particular, Cancer cell 1 (enriched in KP^loss^-Late tumors) exhibited high expression levels of classical subtype genes such as *Lgals4*, *Dmbt1*, *Agr2*, *Tff2*, and *Vsig2*. Cancer cell 2 expressed high levels of mesenchymal/basal-like subtype genes including *Lgals1*, *Fbln2*, *Timp1*, and *Areg*. Furthermore, GSEA revealed differentially enriched pathways of the three cancer cell subtypes. Specifically, Cancer cell 1 subcluster highly expressed the signature genes of Interferon α, Interferon γ, KRAS, and p53 pathways, while the MYC pathway was downregulated. Cancer cell 2 subcluster highly expressed the signature genes of MYC target v1, MYC target v2, and oxidative phosphorylation pathways. Cancer cell 3 subcluster highly expressed the signature genes of Interferon α, Interferon γ, and KRAS pathways, which were also observed in Cancer cell 1 (Appendix A). In summary, these results revealed that the cancer cells of KP^loss^-Late tumors exhibit an enriched epithelial/classical phenotype, which was similar to that of KP^mut^-Early tumors. In contrast, the cancer cells of KP^mut^-Late tumor exhibit enriched subcluster with mesenchymal/basal-like phenotype.

### 3.3. Tumors with p53 Mutation Reveal Elevated Expression of HMGA2 in Cancer Cells as Compared to Tumors with p53 Loss

Further analysis on cancer cell-specific genes revealed that KP^mut^-Late tumors exhibit elevated expression levels of several genes, including *Prkg2*, *Hmga2*, *Wincr1*, *Cdkn2a*, *Tnnt2*, *Aqp5*, and *Sprr1* (Figure 3A). Notably, these genes were upregulated specifically in cancer cells (Appendix A). Next, we utilized the TCGA database to compare the gene profiles of human PDAC with p53 mutation or p53 loss. We selected those patient samples with KRAS G12 mutations and then stratified them into two groups (with either p53 mutation or p53 loss). Based on the genetic alterations of *TP53*, the gain-of-function mutations were categorized as p53 mutation (*n* = 23), while the loss-of-function and truncating mutations were classified as p53 loss (*n* = 25) (Figure 3B). Further analysis of the TCGA pancreatic cancer dataset revealed that p53 gain-of-function mutations were associated with moderately worse survival of patients, as compared with p53 loss-of-function mutations (Appendix A). The expression of *HMGA2* in the TCGA database and the expression of *Hmga2* in our single-cell RNA-sequencing data were consistently upregulated in p53 mutant tumors, as compared to p53 loss tumors (Figure 3C,D). As expected, we also observed a significant decrease of *TP53* expression in the p53 loss cases as compared to p53 mutation cases (Figure 3D). These findings suggested the distinct genetic variances of p53 mutation and p53 loss might impact the phenotypes of both the cancer cells and their microenvironment.

### 3.4. Compositions of B Cells and T Cells Vary among KP^mut^-Early, KP^mut^-Late, and KP^loss^-Late Tumors

According to our previous results, p53 mutation and loss may differentially regulate the phenotypes of cancer cells and the tumor microenvironment. Therefore, we further compared the alterations in immune cell profiles between KP^mut^ tumors and KP^loss^ tumors. We observed that the B cells in KP^mut^ tumors exhibited different phenotypes from those in KP^loss^ tumors (Figure 4A). We divided B cells into Pro-B cell, Pre-B cell, Pre-BCR, Naive B cell, Germinal center B cell, and Plasma cell based on the reported signature genes (Figure 4A,B) [33,34,35,36,37,38]. Subsequently, the B cell compositions of the KP^mut^-Early, KP^mut^-Late, and KP^loss^-Late groups were analyzed (Figure 4C). We observed that KP^mut^-Late tumors had higher numbers of Pro-B cells, Pre-B cells, and Pre-BCR than KP^loss^-Late tumors (or KP^mut^-Early tumors). Pro-B cells, Pre-B cells, and Pre-BCR were hardly detectable in KP^loss^-Late tumors or KP^mut^-Early tumors. Notably, all three groups of mice contained a considerable amount of naïve B cells (Figure 4C). Upon further analysis, we discovered distinct states of naïve B cells in three types of mice: naïve B cell 0, naïve B cell 1, and naïve B cell 2, present in the KP^mut^-Early, KP^mut^-Late, and KP^loss^-Late groups, respectively (Figure 4D,E). Through extensive GSEA analysis, we identified 28 overlapping pathways among these naïve B cell subtypes. Subsequent differential analysis revealed the similarities between naïve B cell 0 and naïve B cell 2 (Figure 4F,G). These results indicated that the expression profile of B cells in KP^loss^-Late tumors is more analogous to that in KP^mut^-Early tumors, which is distinct from that in KP^loss^-Late tumors.

We also analyzed the cell composition of T cell subclusters (Figure 5A,B). In general, the T cell population of KP^mut^-Late tumors was predominantly composed of regulatory T cells (Tregs; FOXP3+CD4+), with a small proportion of NK cells. In contrast, the T cell population in KP^loss^-Late tumors was primarily composed of CD4+FOXP3 T cells, followed by Tregs and CD8+ T cells. Interestingly, the numbers of innate lymphoid cell subclusters, ILC2 and ILC3, increased in KP^mut^-Late tumors, which merits further investigation to determine the underlying mechanism. To summarize, T cells revealed distinct subtype composition patterns between KP^mut^-Late and KP^loss^-Late tumors.

### 3.5. Neutrophil Subpopulations Differ among KP^mut^-Early, KP^mut^-Late, and KP^loss^-Late Tumors

We next analyzed the differences in neutrophils across KP^mut^-Early, KP^mut^-Late, and KP^loss^-Late tumors. We stratified the total granulocyte/neutrophil population into three subclusters: Gran-Mo progenitor, G-MDSC, and Mature neutrophil (Figure 6A,B). We observed that KP^mut^-Late tumors exhibited the presence of Gran-Mo progenitor, G-MDSC, and Mature neutrophil. In contrast, KP^loss^-Late tumors and KP^mut^-Early tumors exhibited a predominant presence of G-MDSC (Figure 6C,D). These findings provide valuable insights into the cellular components of the samples and can aid in further research endeavors. Based on a recently published methodology [24], we analyzed the signature scores of neutrophils in tumors. We observed that the immunosuppression signature of tumor-associated neutrophils was significantly upregulated in KP^loss^-Late tumors but was downregulated in KP^mut^-Late tumors (Figure 6E,F). Notably, KP^loss^-Late tumors exhibited similar profiles to KP^mut^-Early tumors in terms of pathways including angiogenesis, ECM remodeling, myeloid cell recruitment, proliferation, and neutrophil cytotoxicity, while KP^mut^-Late tumors revealed downregulation of several of the aforementioned pathways (Appendix A).

### 3.6. KP^mut^-Early, KP^mut^-Late, and KP^loss^-Late Tumors Reveal Different Compositions of Myeloid Cell and Dendritic Cell (DC) Subclusters

Next, we plotted the signature genes of myeloid cells and DCs (Figure 7A,B). Myeloid cells could be categorized into four subclusters: myeloid cell 1, myeloid cell 2, myeloid cell 3, and myeloid cell 4. Meanwhile, DCs were divided into cDC1, cDC2, and pDC. Notably, myeloid cell 1 exhibited higher levels of *H2-Aa*, *H2-Ab1*, and *Cd74*, while myeloid cell 2 displayed elevated expression of *Retnla* and *Fcrls*. Myeloid cell 3 presented higher levels of *Arg1*, *Hilpda*, *Serpine1*, *Rnase2a*, *Chil3*, while myeloid cell 4 had lower levels of *Csf1r*, *Mrc1*, *C1qa*, *C1qb*, and *C1qc*, but highly expressed *Hp*, *Ifitm6*, *Gpr141*, and *Ms4a4c*. Moreover, both myeloid cell 3 and myeloid cell 4 expressed *Mmp8* and *Vcan*. Lastly, cDCs were primarily divided into cDC1 with high expression of *Naaa* and cDC2 with high expression of *Ccl22*/*Ccl5* (Appendix A). We observed a higher proportion of Myeloid cell 3 in both KP^mut^-Late and KP^loss^-Late (Figure 7C). GSEA identified that myeloid cell 3 subcluster has upregulated pathways including Oxidative_phosphorylation, Glycolysis, Fatty acid metabolism, mTORC1 signaling, Unfolded protein response, Protein secretion, Adipogenesis, TNFA signaling via NF-kB (Figure 7D and Appendix A). Further analysis showed that glycolysis in myeloid cell 3 has a strong positive correlation with both oxidative phosphorylation and hypoxia (Figure 7E,F and Appendix A). Recent studies reported that monocytes can simultaneously increase the efficiency of glycolysis and oxidative phosphorylation under the stimulation of low concentrations of LPS [39]. However, the exact role of these cellular functions in tumors warrants further exploration.

### 3.7. Compositions of Endothelial Cells and Fibroblasts Vary among KP^mut^-Early, KP^mut^-Late and KP^loss^-Late

We observed two distinct subpopulations of endothelial cells that result from p53 mutation and p53 loss, known as endothelial-1 (EC-1) and endothelial-2 (EC-2) (Figure 8A). EC-1 was primarily found in KP^mut^-Early and KP^mut^-Late with the expression of *Apoe*, *Lyz2*, and *Cd52*, while EC-2 was mainly found in KP^loss^-Late with the expression of *Mmp14*, *Ecm1*, *Col13a*, and *Lamb* (Figure 8B,C).

Next, we analyzed the compositions and related characteristics of cancer-associated fibroblasts (CAFs) KP^mut^-Early, KP^mut^-Late, and KP^loss^-Late tumors. CAFs were categorized based on specific marker genes [5] into three subtypes: inflammatory CAF (iCAF), myofibroblast (myCAF), and antigen-presenting CAF (apCAF) (Figure 8D,E). We observed that iCAF was predominantly present in KP^mut^-Early tumors, while myCAF and apCAF subtypes were enriched in KP^mut^-Late tumors (Figure 8F). The CAF subcluster composition between KP^loss^-Late tumors and KP^mut^-Late tumors was similar, although the total number of CAFs in KP^loss^-Late tumors was significantly lower than that in KP^mut^-Late tumors (Figure 8F). Taken together, our findings indicated that p53 loss and p53 mutation in pancreatic cancer cells differentially affect the subclusters of various cell populations such as endothelial cell, CAFs, and immune cells.

## 4. Discussion

Pancreatic cancer is one of the deadliest cancers worldwide and is refractory to most therapies, due to the complex nature of its tumor microenvironment [1]. As a result, extensive investigations are still required to elucidate the biology of the tumor microenvironment of pancreatic cancer. There are several oncogenic drivers that contribute to the development of pancreatic ductal adenocarcinoma (PDAC), including KRAS and p53 mutations. While oncogenic KRAS mutation is dominated by the KRAS G12* format in PDAC, p53 mutations can have either gain-of-function or loss-of-function alterations. Previous studies have shown that both gain-of-function or loss-of-function types of p53 mutations can occur in human pancreatic cancer and have distinct impacts on tumor progression [15,16]. Oncogenic effects are often observed in the gain-of-function p53 mutations, which are commonly found in R175H, R248Q, R248W, R273C, R273H, G245S, H179R, Y163C, Y220C, Y234C, G249, and R282 [19,20]. On the other hand, loss-of-function p53 mutations are associated with increased tumor penetrance and decreased life expectancy, as seen in the context of the p53 E224D mutant [40]. A recent study suggested that p53 loss induces ordered and deterministic cancer genome evolution [41]. Another study previously reported that the genotype plays a crucial role in modulating the tumor microenvironment of PDAC, leading to the induction of matricellular fibrosis and the progression of the tumor [42]. It is crucial to investigate the distinct effects induced by these oncogenic mutations on both cancer cells and the surrounding microenvironment, which may reveal potential vulnerabilities of PDAC that are associated with specific cancer genotypes.

Previous studies also demonstrated that gain-of-function p53 mutations can have significantly different oncogenic effects as compared to the loss-of-function p53 mutations [15,16]. Utilizing a single-cell technique and various transgenic mouse models, we can conduct an in-depth comparison of the tumor microenvironment components between p53-mutated and p53-loss pancreatic tumors, both in the context of oncogenic KRAS^G12D^-driven PDAC. Our study focuses on identifying the transcriptomic alterations in various cell populations across early-stage p53-mutated (KP^mut^-Early) tumors, late-stage p53-mutated (KP^mut^-Late) tumors, and late-stage p53-loss (KP^loss^-Late) tumors.

The utilization of transgenic mouse models has dramatically facilitated the in-depth investigation on the complex biology of PDAC in the past decades. Transgenic mouse models of oncogenic KRAS-driven PDAC with either p53 mutation [43] or p53 loss [44,45,46] have been widely used by numerous studies. Both the p53-mutated *Kras^G12D^*; *Trp53^R172H^* (KP^mut^) mouse model and p53-loss *Kras^G12D^*; *Trp53^cKO^* (KP^loss^) mouse model are clinically relevant and can faithfully recapitulate human PDAC with distinct genotypes. The KP^mut^ mouse model can recapitulate the human PDAC cases with KRAS^G12D^ mutation and p53^R172H^ mutation (or other similar gain-of-function mutations), with 23 cases within the TCGA PDAC cohort as shown in this present study. In comparison, The KP^loss^ mouse model can recapitulate the human PDAC cases with KRAS^G12D^ mutation and p53 loss-of-function mutations/truncations, with 25 cases within the TCGA PDAC cohort. The p53 loss-of-function mutations and truncations have been widely observed in various cancer types [47,48,49]. The previous findings are consistent with our results showing that patients with p53 gain-of-function mutations exhibit worse prognosis than patients with p53 loss-of-function mutations, based on the analysis of TCGA pancreatic cancer cohort data.

By integrating our recently published single-cell datasets [9,10,11,13], we compared the similarities and differences between p53-mutated (KP^mut^) and p53-loss (KP^loss^) tumors. Our analysis identified the enrichment of a mesenchymal/basal-like cancer cell subcluster in late-stage KP^mut^ tumors, but not in late-stage KP^loss^ tumors or early-stage KP^mut^ tumors. Furthermore, our study has revealed that HMGA2 is upregulated in both mouse PDAC tumors of KP^mut^-Late mice and human PDAC samples with p53 mutations in TCGA dataset (p53 mutation group, *n* = 23; as compared to p53 loss group, *n* = 25). It has been reported that HMGA2 exhibits a strong association with tumor metastasis and growth [50]. Besides the Cancer cell 2 subpopulation uniquely enriched in KP^loss^ tumors, we also identified a shared Cancer cell 1 subpopulation that is abundantly present in both KP^loss^ tumors and KP^mut^ tumors. These results indicate that this epithelial/classical Cancer cell 1 subpopulation is presumably the common cancer-initiating cell lineage during PDAC development.

Also, we observed similarities of many immune cell populations between KP^mut^ and KP^loss^ tumors. In fact, the overall cell population compositions and phenotypes were surprisingly similar between KP^mut^ and KP^loss^ mouse models, as revealed by scRNA-seq (Figure 1). These findings support the notion that both KP^mut^ and KP^loss^ mouse models are useful for studying the tumor microenvironment of PDAC, consistent with previous studies using both model systems [17,43,44,45,46]. Specifically, both KP^mut^-Late and KP^loss^-Late tumors exhibit significantly enriched immunosuppressive myeloid cells with high expression levels of *Arg1* and *Chil3*, which resembles the phenotype of tumor-associated macrophage. These results indicate that the enrichment of immunosuppressive myeloid cells is likely a common feature associated with late-stage PDAC, despite the presence of p53 mutation or p53 loss. Despite the similar enrichment of myeloid cells in KP^mut^-Late and KP^loss^-Late tumors, the phenotype and composition of tumor-associated neutrophils in KP^loss^-Late tumors are significantly different from those in KP^mut^-Late tumors. Future studies are required to determine whether leveraging these myeloid cells and neutrophils can provide insights into new therapeutic strategies for PDAC.

## 5. Conclusions

To summarize, we conducted a thorough analysis across multiple datasets of scRNA-seq data to compare the tumor microenvironment of oncogenic KRAS-driven PDAC harboring p53 mutation or p53 loss. Our analyses uncovered distinctive phenotypic and transcriptional profiles of various cell populations in the context of p53 mutation as compared to p53 loss. We also identified the similarities in overall cell type compositions between KP^mut^ and KP^loss^ mouse models. While this study provides valuable insights, further investigations are necessary to fully comprehend the mechanisms that govern the distinct roles of p53 mutation and p53 loss in regulating the tumor microenvironment and PDAC progression.

## Figures and Tables

**Figure 1 cells-12-02614-f001:**
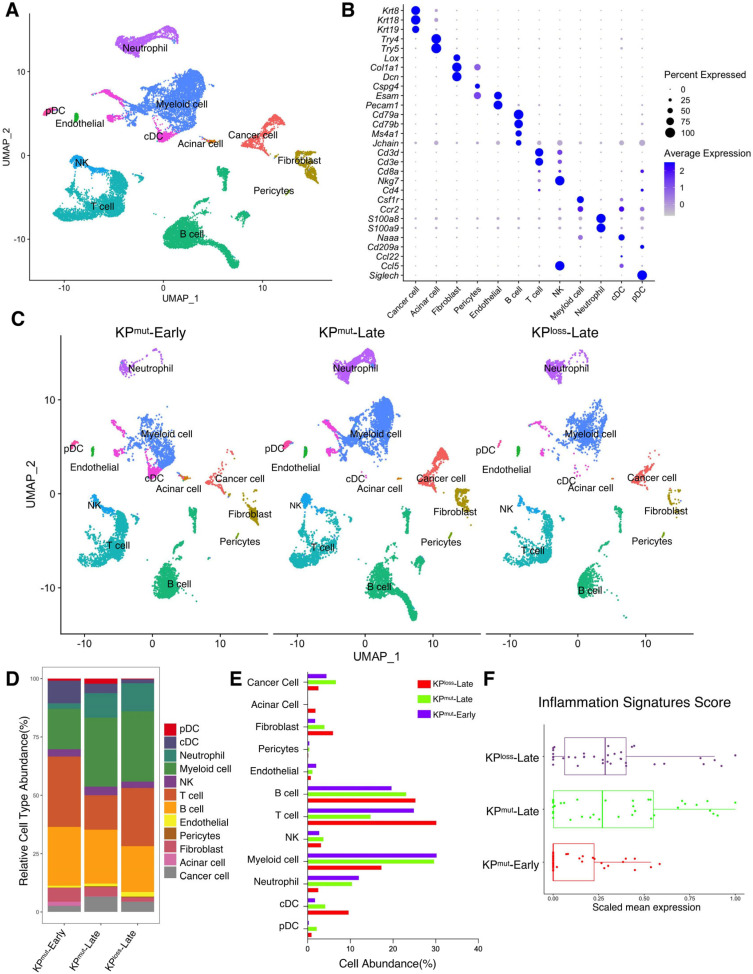
Single-cell RNA-sequencing (scRNA-seq) analysis identifies distinct compositions of cell populations in oncogenic KRAS-driven pancreatic tumors with different p53 status. (**A**,**B**) scRNA-seq analysis of unfractionated live cell mixture from oncogenic KRAS^G12D^-driven PDAC tumors with either p53 mutation (KP^mut^ group) or p53 loss (KP^loss^ group), as published in our previous datasets (GSE198815 and GSE166298). The major cell clusters are shown in a UMAP plot (**A**). (**B**) Dot plot showing the normalized expression of pre-defined marker genes for defined cell clusters. (**C**) UMAP plot that compares the cellular compositions of three groups of transgene mice (*n* = 3 per group): early-stage primary tumors (KP^mut^-Early), late-stage primary tumors (KP^mut^-Late), and late-stage p53 loss tumors (KP^loss^-Late). (**D**) Cell-type abundance of all captured cells across KP^mut^-Early, KP^mut^-Late, and KP^loss^-Late groups. (**E**) Abundance (%) of cell types across KP^mut^-Early, KP^mut^-Late, and KP^loss^-Late groups. (**F**) Scaled mean expression of inflammation signatures (*n* = 38) in cells from various groups. The median (horizontal line), second to third quartiles (box), and Tukey-style whiskers (beyond the box) are represented by the boxes. The dots denote the individual signatures. cDC, conventional dendritic cell; pDC, plasmacytoid dendritic cell; G-MDSC, granulocytic myeloid-derived suppressor cell; Gran-Mo progenitor, granulocyte-monocyte progenitor cell.

**Figure 2 cells-12-02614-f002:**
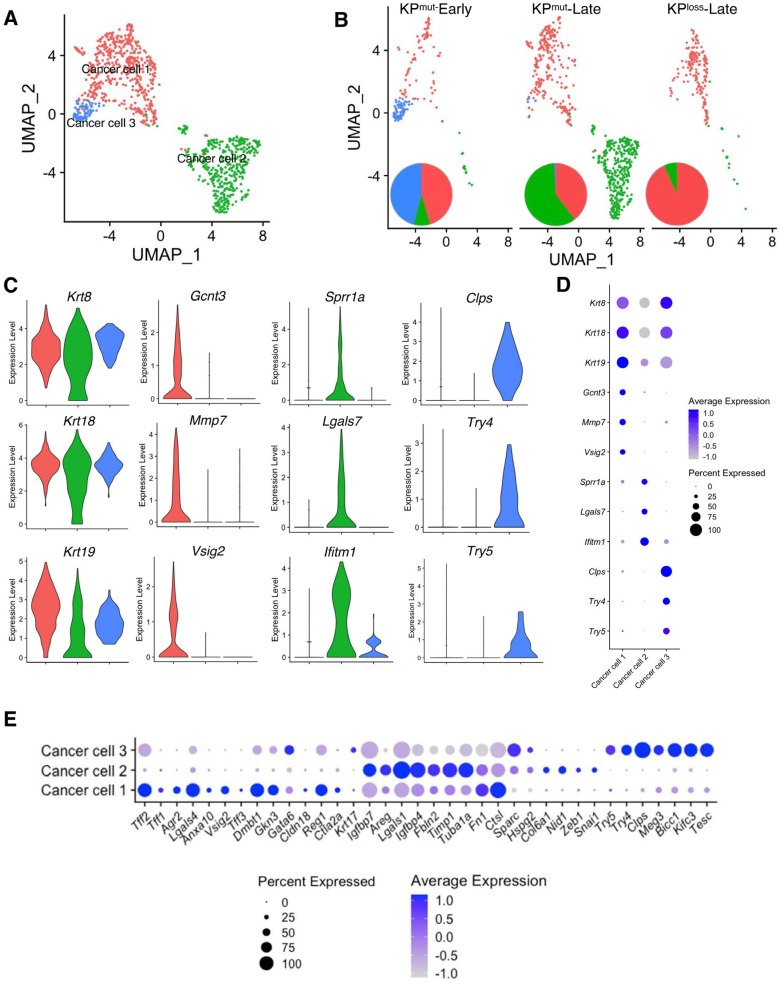
Cancer cell subpopulations are different between KP^mut^ and KP^loss^ pancreatic tumors. (**A**,**B**) Cancer cells from KP^mut^-Early, KP^mut^-Late, and KP^loss^-Late groups are stratified into three distinct subclusters in the UMAP plot (**A**) and compared across three groups (**B**). The pie charts also show the changes in cancer cell subcluster composition among the three groups. (**C**,**D**) Expression profiles of signature genes for the cancer cell subpopulations, shown as violin plots (**C**) and dot plots (**D**). (**E**) The cancer cell subpopulations were examined for expression profiles of genes associated with the definition of classical and basal-like cancer cell subtypes, as shown in dot plots. Acinar-like signature genes are also shown in comparison with classical and basal-like signature genes.

**Figure 3 cells-12-02614-f003:**
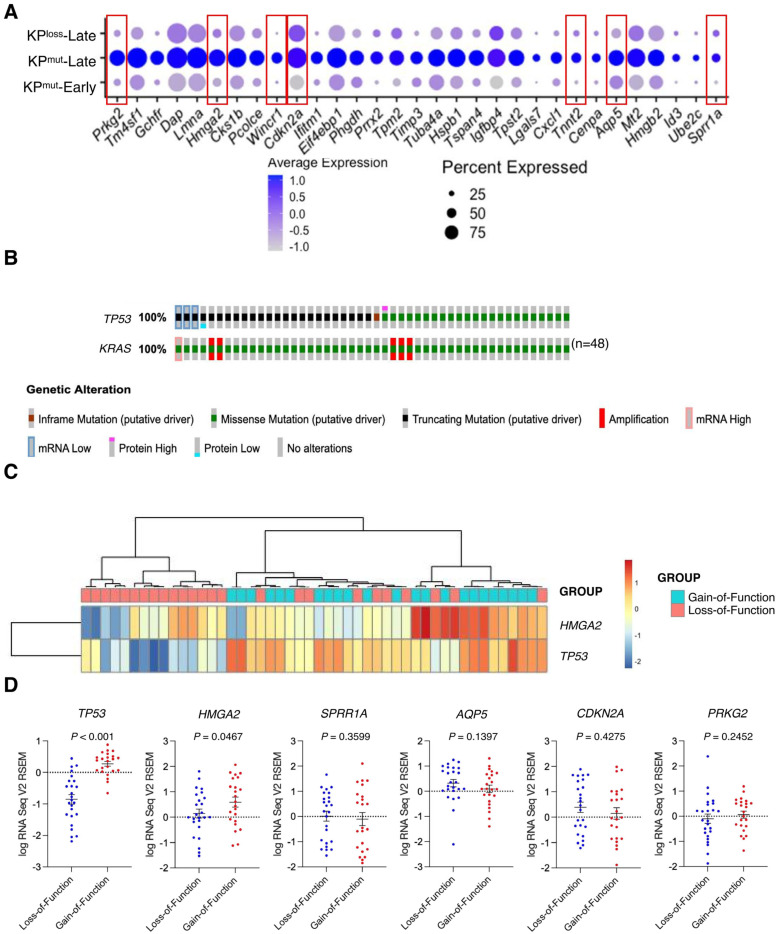
Differentially expressed genes (DEGs) are identified by comparing cancer cells from KP^mut^-Late with cancer cells from KP^mut^-Early and KP^loss^-Late groups. (**A**) Expression profiles of indicated marker genes in cancer cells from various groups are shown in a dot plot. Cancer cell-specific genes with elevated expression levels in KP^mut^-Late group were indicated by red boxes. (**B**) Samples from TCGA pancreatic cancer cohort with indicated (either loss-of-function or gain-of-function) *TP53* mutations together with *KRAS* G12* mutations. (**C**) Heatmap showing the expression of selected marker genes in PDAC samples with loss-of-function or gain-of-function p53 mutations from the TCGA pancreatic cancer cohort. (**D**) Expression profiles of selected marker genes in PDAC samples with loss-of-function or gain-of-function p53 mutations. Statistical differences between groups in panel (**D**) were assessed by *t*-test.

**Figure 4 cells-12-02614-f004:**
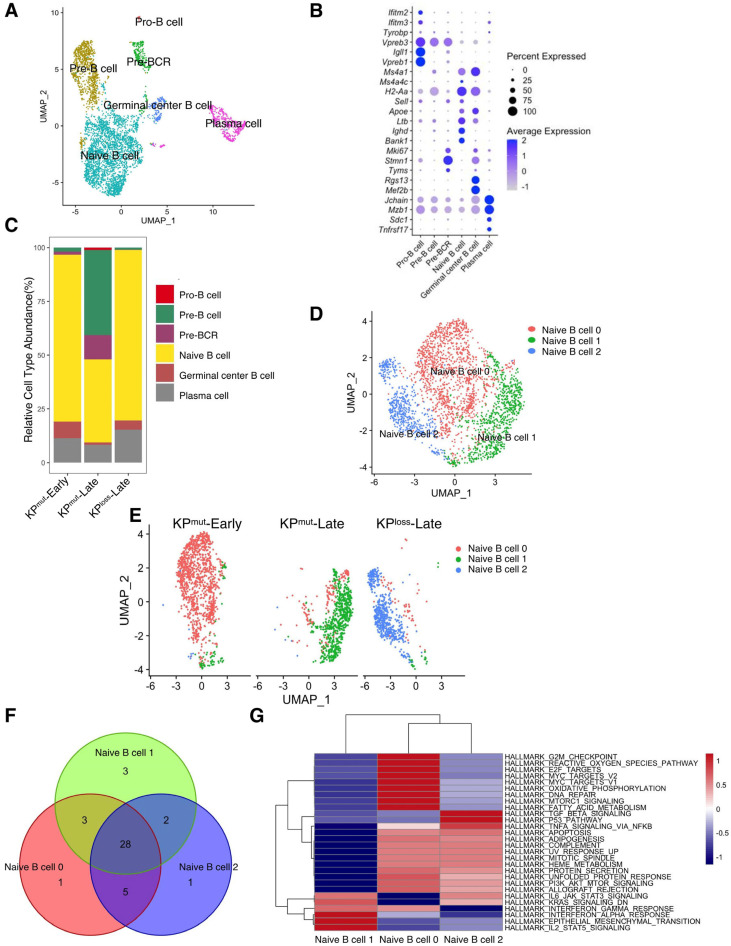
Comparing B cell subtype compositions in KP^mut^-Early, KP^mut^-Late, and KP^loss^-Late tumors. (**A**,**B**) B cells from KP^mut^-Early, KP^mut^-Late, and KP^loss^-Late tumors are segregated into distinct subclusters in the UMAP plot (**A**) based on the diverse expression profiles of signature genes depicted in the dot plot (**B**). (**C**) B cell-subtype abundance across KP^mut^-Early, KP^mut^-Late, and KP^loss^-Late groups. (**D**) Naïve B cells from KP^mut^-Early, KP^mut^-Late, and KP^loss^-Late are segregated into distinct subclusters in the UMAP plot. (**E**) UMAP plot comparing the Naïve B cell compositions of three groups. (**F**) Venn diagram of the GSEA across naive B cell subtypes. (**G**) Heatmap of the GSEA across three Naïve B cell subtypes.

**Figure 5 cells-12-02614-f005:**
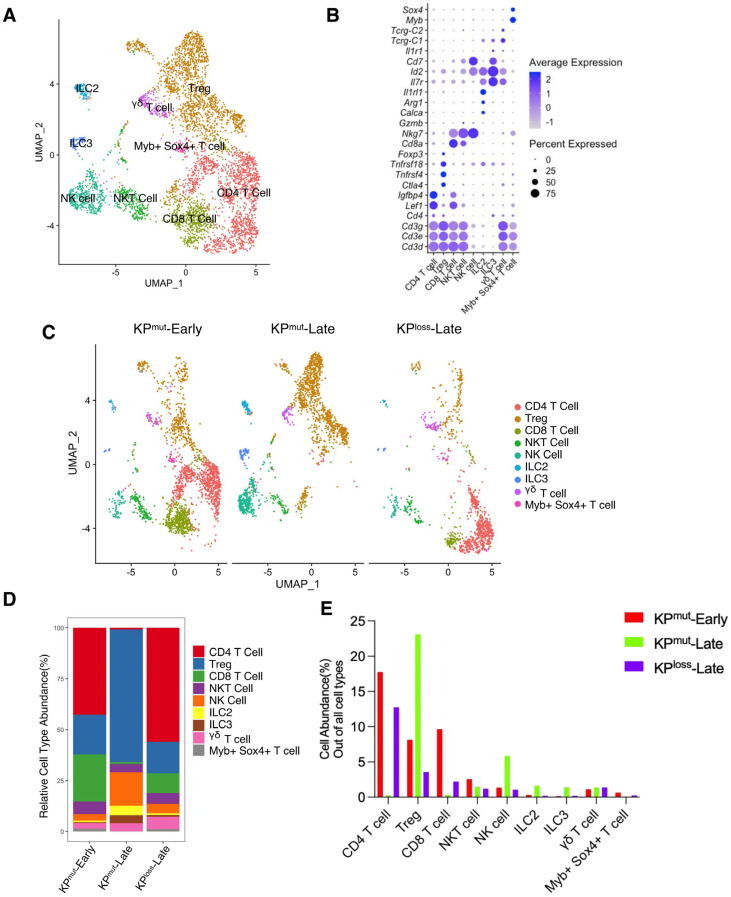
Subpopulation compositions of T cells and NK cells in KP^mut^-Early, KP^mut^-Late, and KP^loss^-Late tumors. (**A**,**B**) T cells and NK cells from KP^mut^-Early, KP^mut^-Late, and KP^loss^-Late tumors are classified into distinct subclusters in the UMAP plot (**A**) based on the diverse expression profiles of signature genes depicted in the dot plot (**B**). (**C**) UMAP plot comparing the T cells and NK cells compositions across three groups. (**D**) T and NK cell subtype abundance across KP^mut^-Early, KP^mut^-Late, and KP^loss^-Late groups. (**E**) abundance (%) of T and NK cells across KP^mut^-Early, KP^mut^-Late, and KP^loss^-Late groups.

**Figure 6 cells-12-02614-f006:**
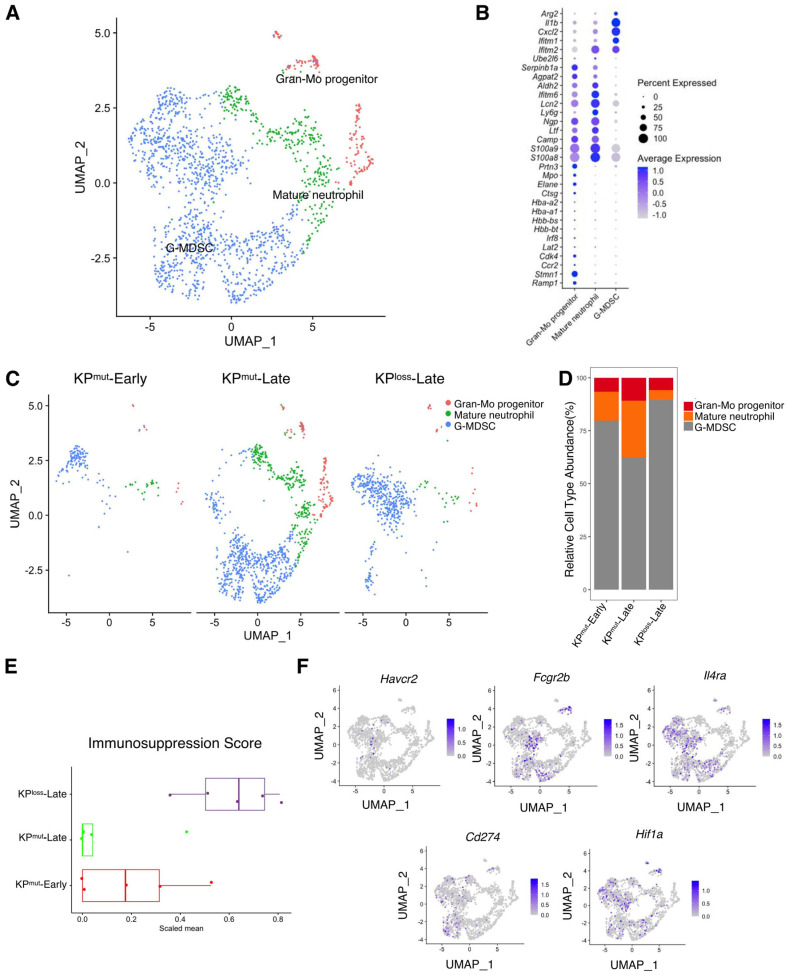
Similarities and differences of neutrophil subclusters across KP^mut^-Early, KP^mut^-Late, and KP^loss^-Late tumors. (**A**,**B**) Neutrophils from KP^mut^-Early, KP^mut^-Late, and KP^loss^-Late tumors are segregated into distinct subclusters in the UMAP plot (**A**) based on the diverse expression profiles of signature genes depicted in the dot plot (**B**). (**C**) The UMAP plot compares the neutrophil compositions across KP^mut^-Early, KP^mut^-Late, and KP^loss^-Late groups. (**D**) Neutrophil subtype abundance across KP^mut^-Early, KP^mut^-Late, and KP^loss^-Late groups. (**E**) Scaled mean expression of immunosuppression signatures (*n* = 5) in cells from various groups. The median (horizontal line), second to third quartiles (box), and Tukey-style whiskers (beyond the box) are represented by the boxes. The dots denote the individual signatures. (**F**) Immunosuppression-related genes (*Havcr2*, *Fcgr2b*, *Il4ra*, *Cd274*, *Hif1a*) are characterized in a UMAP plot.

**Figure 7 cells-12-02614-f007:**
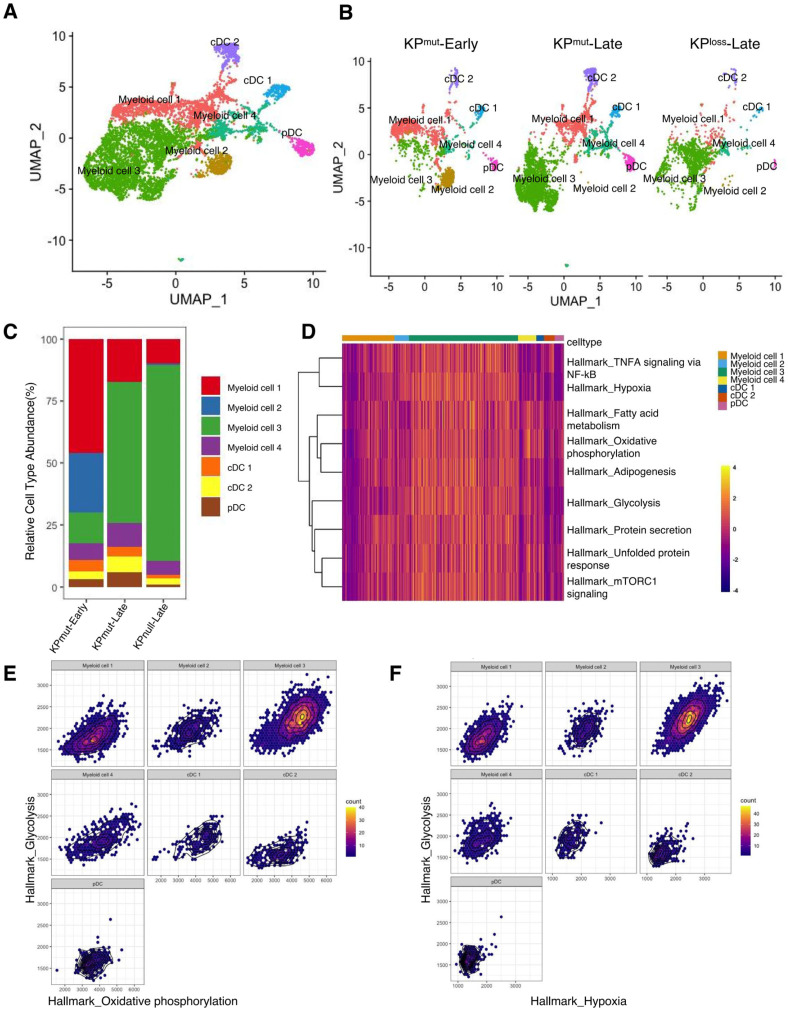
Compositions of myeloid cells and dendritic cells (DCs) in KP^mut^-Early, KP^mut^-Late, and KP^loss^-Late tumors. (**A**,**B**) Myeloid cells and DCs from KP^mut^-Early, KP^mut^-Late, and KP^loss^-Late tumors are segregated into distinct subclusters in the UMAP plot(**A**) and compared across the groups (**B**). (**C**) Myeloid cell and DC subtype abundance across KP^mut^-Early, KP^mut^-Late, and KP^loss^-Late groups. (**D**) Heatmap showing the expression profile of ‘Myeloid cell 3’-associated GSEA pathways in myeloid cell and DC subpopulations. (**E**) Relationship between Hallmark_glycolysis and Hallmark_oxidative_phosphorylation examined in myeloid cell and DC subtypes. (**F**) Correlation between Hallmark_glycolysis and Hallmark_hypoxia in myeloid cell and DC subtypes.

**Figure 8 cells-12-02614-f008:**
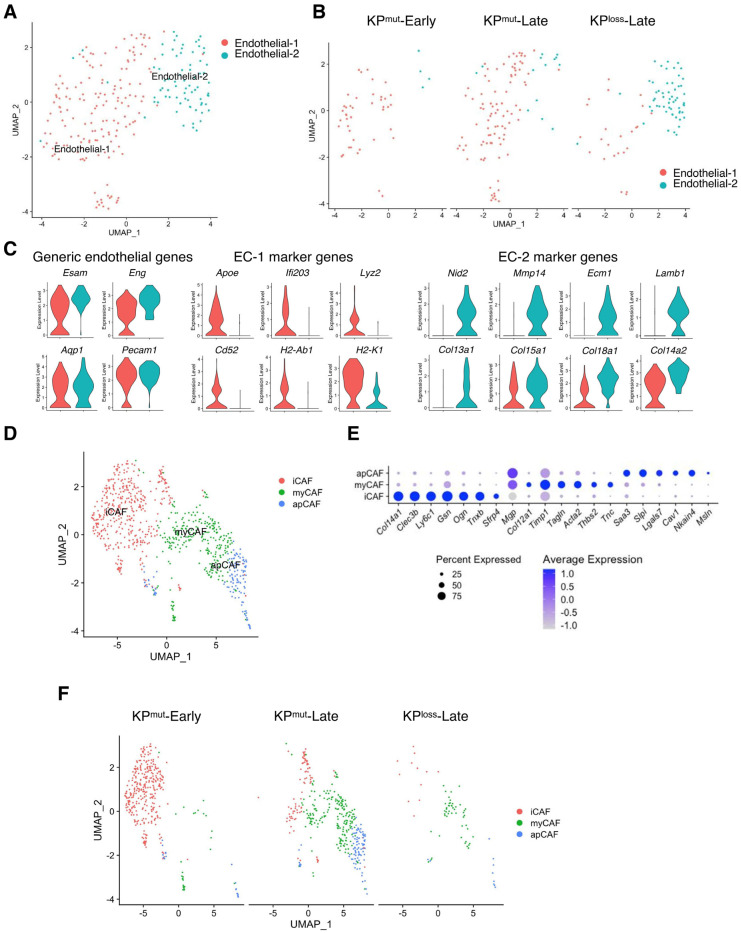
Subclusters of endothelial cells and cancer-associated fibroblasts (CAFs) in KP^mut^-Early, KP^mut^-Late, and KP^loss^-Late tumors. (**A**,**B**) Endothelial cells from KP^mut^-Early, KP^mut^-Late, and KP^loss^-Late tumors are segregated into distinct subclusters in the UMAP plot (**A**) and compared across the groups (**B**). (**C**) Signature genes of endothelial cells from KP^mut^-Early, KP^mut^-Late, and KP^loss^-Late tumors shown in a violin plot. (**D**,**E**) Inflammatory CAFs (iCAF), myofibroblastic CAFs (myCAF), and antigen-presenting CAF (apCAF) subclusters are denoted. CAFs obtained from KP^mut^-Early, KP^mut^-Late, and KP^loss^-Late tumors are segregated into aforementioned subclusters in the UMAP plot (**D**) based on the diverse expression profiles of signature genes depicted in the dot plot (**E**). (**F**) UMAP plot comparing the CAF subtype compositions across three groups.

## Data Availability

All datasets of transgenic mouse models analyzed in this study were previously deposited at the National Center for Biotechnology Information’s Gene Expression Omnibus (GEO) database repository with the following accession numbers: GSE198815 and GSE166298. The survival and gene expression data of TCGA pancreatic adenocarcinoma cohort were based on the GDAC Firehose PAAD dataset.

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
