# Peer review of "Single-Cell Analysis Differentiates the Effects of p53 Mutation and p53 Loss on Cell Compositions of Oncogenic Kras-Driven Pancreatic Cancer"

_cells, 2023, doi:10.3390/cells12222614_

Round 1

Reviewer 1 Report

Comments and Suggestions for Authors

This article follows up on previous data obtained in the authors' lab providing an extensive analysis of p53 activating mutation and p53 on K-Ras pancreatic cancer. The data are thorough and provides novel insights into the different effects of both p53 loss-of function and p53 gain-of function in pancreatic cancer, which should be of interest to a very wide audience. I am thus in principle willing to accept this manuscript for publication. My minor concerns would be regarding clarity:

- The title could be improved as it does not reflect any of the positive observations described in the manuscript, and it is mainly descriptive of what is being done.

- The introduction is quite short and there is little in it regarding p53. Importantly, the authors do not describe what the R172H mutation in p53 does.

- The use of the term p53 mutation throughout is vague, given that mutations can be either activating or inactivating. Could the authors perhaps refer to their p53 mutation as either a dominant positive p53 mutation, p53 activating mutation, constitutively active p53 mutation or others?

Author Response

This article follows up on previous data obtained in the authors' lab providing an extensive analysis of p53 activating mutation and p53 on K-Ras pancreatic cancer. The data are thorough and provides novel insights into the different effects of both p53 loss-of function and p53 gain-of function in pancreatic cancer, which should be of interest to a very wide audience. I am thus in principle willing to accept this manuscript for publication. My minor concerns would be regarding clarity:

Response: We greatly appreciate the reviewer’s overall enthusiasm and insightful comments. According to the suggestions, we have carefully revised our manuscript to address all the important points.

- The title could be improved as it does not reflect any of the positive observations described in the manuscript, and it is mainly descriptive of what is being done.

Response: Thank you for this important point. As suggested, we have revised the title of this study to better reflect the main conclusions.

- The introduction is quite short and there is little in it regarding p53. Importantly, the authors do not describe what the R172H mutation in p53 does.

Response: Thank you for this insightful suggestion. We completely agree that more introduction and discussion are critically needed in our manuscript. Accordingly, we have added more texts to review the functions of p53R172H in the context of KRAS-driven pancreatic cancer, as added in the Introduction section.

- The use of the term p53 mutation throughout is vague, given that mutations can be either activating or inactivating. Could the authors perhaps refer to their p53 mutation as either a dominant positive p53 mutation, p53 activating mutation, constitutively active p53 mutation or others?

Response: Thank you for this important comment. As suggested, we added more clarification and definition in the Introduction section to define the term ‘p53-mutation’ used in our current study.

Reviewer 2 Report

Comments and Suggestions for Authors

I have now generally read the paper entitled: Single-cell analysis compares the effects of p53 mutation and p53 loss on oncogenic KRAS-driven pancreatic cancer.

I can conclude that the research is vitally important to solve some of the complex nature of PC tumor microenvironment and different types of PC. The work is highly scientific and might at the end benefits the patients prognosis.

Single-cell analysis reveals unique differences in the composition and genomic profile of cell populations between the KPmut and KPloss pancreatic tumors.

The authors suggested the distinct genetic variances of p53 mutation and p53 loss that might impact the phenotypes of both the cancer cells and their microenvironment.

The figures are too many and hard to follow due to the small size.

What is important is to elucidate the mechanisms that govern the distinct roles of p53 mutation and p53 loss in regulating the tumor microenvironment and PDAC progression. Finally, I would like the authors to comment and predict the advantages and disadvantages to PC patients considering the 2 types of P53 gain of function or loss of function.

Comments on the Quality of English Language

English is fine and perhaps minor typo errors have to be checked.

Author Response

I have now generally read the paper entitled: Single-cell analysis compares the effects of p53 mutation and p53 loss on oncogenic KRAS-driven pancreatic cancer.

I can conclude that the research is vitally important to solve some of the complex nature of PC tumor microenvironment and different types of PC. The work is highly scientific and might at the end benefits the patients prognosis.

Response: We sincerely appreciate the reviewer’s overall enthusiasm and insightful comments. According to the suggestions, we have carefully revised our manuscript to address all the important points.

Single-cell analysis reveals unique differences in the composition and genomic profile of cell populations between the KPmut and KPloss pancreatic tumors.

The authors suggested the distinct genetic variances of p53 mutation and p53 loss that might impact the phenotypes of both the cancer cells and their microenvironment.

The figures are too many and hard to follow due to the small size.

Response: We apologize for the fact that there are too many figures and supplementary figures, as we intended to thoroughly present our single-cell analysis data. In order to improve the readability of our figures, we have modified the label/size of many figures and removed a few non-essential supplementary figures.

What is important is to elucidate the mechanisms that govern the distinct roles of p53 mutation and p53 loss in regulating the tumor microenvironment and PDAC progression. Finally, I would like the authors to comment and predict the advantages and disadvantages to PC patients considering the 2 types of P53 gain of function or loss of function.

Response: Thank you for this important point. We have added new results by analyzing TCGA pancreatic cancer dataset (new Supplementary Figure 3B) to compare the patient outcomes between p53 gain-of-function and loss-of-function groups. We also added more discussions on this point accordingly.